# Implications of Mineralization Biomarkers in Breast Cancer Outcomes Beyond Calcifications

**DOI:** 10.3390/ijms26020645

**Published:** 2025-01-14

**Authors:** Erica Giacobbi, Rita Bonfiglio, Gabriele Rotondaro, Francesca Servadei, Artem Smirnov, Valeria Palumbo, Maria Paola Scioli, Elena Bonanno, Claudio Oreste Buonomo, Gianluca Vanni, Eleonora Candi, Alessandro Mauriello, Manuel Scimeca

**Affiliations:** 1Department of Experimental Medicine, TOR, University of Rome “Tor Vergata”, 00133 Rome, Italy; erica.giacobbi@gmail.com (E.G.); rita.bonfiglio@uniroma2.it (R.B.); gabrielerotondaro@gmail.com (G.R.); francescaservadei@gmail.com (F.S.); artem.smirnov@uniroma2.it (A.S.); valeria.palumbo95@hotmail.com (V.P.); sciolimp@hotmail.it (M.P.S.); elena.bonanno@uniroma2.it (E.B.); candi@uniroma2.it (E.C.); manuel.scimeca@uniroma2.it (M.S.); 2Breast Unit, Department of Surgical Science, University of Rome “Tor Vergata”, 00133 Rome, Italy; o.buonomo@inwind.it (C.O.B.); vanni_gianluca@yahoo.it (G.V.)

**Keywords:** breast cancer, biomarkers, microcalcifications, EMT, BMP-2, BMP-4, SDF-1, vimentin

## Abstract

The aim of this work was to explore the biomarkers associated with epithelial to mesenchymal transition (EMT) and mineralization processes as new prognostic factors across different breast cancer phenotypes. To this end, 133 breast biopsies, including benign and malignant lesions, with or without microcalcifications, were retrospectively collected. Immunohistochemical analysis was performed to evaluate the expression of vimentin, BMP-2, BMP-4, RANKL, Runx2, OPN, PTX3, and SDF-1, while Kaplan—Meier plots were used to assess their prognostic impact on overall survival in a dataset of 2976 breast cancer patients. The expression of vimentin, BMP-2, BMP-4, and SDF-1 was significantly higher in malignant lesions compared to benign ones, regardless of the presence of microcalcifications. Notably, these markers showed no correlation with traditional prognostic factors, such as tumor grade or hormone receptor status. The bioinformatics analysis provided valuable insights into the possible prognostic and therapeutic significance of BMP-2, BMP-4, SDF-1, and vimentin in breast cancer. In fact, all these biomarkers impact on the overall survival in specific molecular breast cancer types. In addition, high expression of SDF-1 and vimentin is able to predict the response to chemotherapy. The findings here reported suggest that vimentin, BMP-2, BMP-4, and SDF-1 could be independent prognostic biomarkers in breast cancer, providing insights beyond traditional clinical factors.

## 1. Introduction

Breast cancer stands as the most common malignancy affecting women worldwide and remains a leading cause of cancer-related deaths, despite significant advancements in diagnosis, treatment, and awareness campaigns [1]. Specifically, it remains the most common malignancy among women worldwide, with almost 2.3 million new diagnoses each year, representing approximately 11.5% of all cancer cases globally [2]. Despite advancements in detection and treatment, breast cancer continues to be a leading cause of cancer-related mortality, accounting for more than 600,000 deaths each year. These epidemiological data underscore the urgent need for improved diagnostic and prognostic tools. Traditional prognostic factors, such as tumor size, lymph node involvement, hormone receptor status, and HER2 expression, have been instrumental in guiding treatment decisions [3,4,5,6]. However, these factors often fail to capture the complex biological heterogeneity of breast cancer. Thus, the management of breast cancer patients requires a multidisciplinary strategy, incorporating regional control measures, such as surgery and radiation, alongside systemic treatments, like endocrine therapy, chemotherapy, and targeted biological therapies [7].

The selection of these treatment modalities is guided by the molecular subtype of the tumor, determined through immunohistochemical (IHC) analysis of specific molecular markers [8]. IHC analysis plays a critical dual role in breast cancer care, functioning as both a prognostic tool and a means to guide therapeutic decisions [9,10].

Pathological assessment of breast cancer focuses on identifying alterations in tissue structure and cellular architecture that are indicative of the disease [11]. Key pathological features, including tumor size, grade, and lymph node involvement, have been extensively analyzed to inform treatment decisions and provide prognostic insights, particularly concerning disease-related morbidity and mortality [12,13].

For instance, tumors expressing hormone receptors generally exhibit more favorable characteristics, whereas triple-negative breast cancer(TNBC) cases often present with less favorable features [12]. As a result, clinical guidelines have been established by integrating both IHC findings and pathological characteristics.

Despite these advancements, the prognostic classifiers currently employed in clinical practice often fall short in accounting for the inherent complexity and diversity of breast cancer.

This high degree of breast cancer heterogeneity [14] poses significant challenges to accurately predict outcomes for all patients using standard classification methods.

In this scenario, during the last decade, a big effort was made in the comprehension of the biological and diagnostic value of breast microcalcifications [15,16,17,18].

The occurrence of specific types of ectopic calcification in breast cancers was indeed associated with EMT, a process known to contribute to tumor invasion and metastasis [19,20,21,22,23,24].

The presence of such calcifications, driven by EMT, was then suggested to be predictive of breast cancer occurrence [25]; microcalcifications have been often linked to tumor progression [26].

Nevertheless, recent research has shown that biomarkers typically involved in bone metabolism can also be expressed by epithelial cancer cells regardless of the presence of ectopic calcifications [27,28,29,30]. In this scenario, several molecules involved in both EMT and bone mineralization, such as vimentin, bone morphogenetic proteins (BMPs), RANKL, and SDF-1 have been proposed as possible prognostic biomarkers of breast cancer. These molecules play crucial roles in cancer progression and metastasis, with vimentin being a key marker of EMT [31], a process that facilitates tumor cell invasion and dissemination. Similarly, BMPs, RANKL, and SDF-1 [32,33,34] contribute to bone remodeling, which is particularly relevant in breast cancer, due to its propensity to metastasize to bone. Despite their biological significance, the clinical relevance of these molecules in predicting breast cancer outcomes remains largely unexplored. To date, no large-scale cohort studies have systematically investigated the prognostic value of these biomarkers or their potential as therapeutic targets. This finding suggests that these biomarkers may play broader roles in cancer biology, potentially influencing tumor progression and metastasis beyond their traditional association with calcification processes.

The aim of this work is to explore the biomarkers associated with EMT and mineralization processes as new prognostic factors across different breast cancer phenotypes.

Specifically, in this study, we perform immunohistochemical and bioinformatic analyses to investigate the possible prognostic role of vimentin, SDF-1, BMP-2, BMP-4, PTX3, OPN, RANKL, and Runx2 on a large cohort of breast cancer patients.

## 2. Results

### 2.1. Histology

The study of H&E sections allowed us to classify breast biopsies according to the last WHO directive [12] into benign lesions (BL, n = 32) and malignant lesions (ML, n = 101), encompassing both in situ (n = 34) and invasive carcinomas (n = 67). Benign lesions and ML were further classified according to the presence/absence of microcalcifications. Benign lesions without microcalcifications (BL−, n  =  18) were fibroadenoma (n = 12) and fibrocystic changes (n = 6), whereas among BL presenting with microcalcifications (BL+, n  =  14) were identified fibroadenoma (n = 7), fibrocystic changes (n = 6), and adenosis (n = 1). Among ML lesions without microcalcifications (ML−, n  =  51) were classified as in situ ductal carcinomas (n = 5), invasive breast carcinoma NST (n = 43), and invasive lobular carcinoma (n = 3). Among ML presenting with microcalcifications (ML+, n  =  50) were identified infiltrating ductal carcinomas (n  =  26) and in situ ductal carcinomas (n  =  24).

### 2.2. In Situ Expression of EMT and Mineralization Markers

Immunohistochemical analyses were performed to investigate the expression of EMT and mineralization markers in BL and ML groups. Specifically, the in situ expression of the following biomarkers has been evaluated: vimentin, SDF-1, BMP-2, BMP-4, PTX3, OPN, RANKL, and Runx2.

As expected, a significant increase in the number of positive breast cells for all investigated biomarkers was observed in the ML group, as compared to the BL ones (Figure 1A).

Vimentin and SDF-1 were the biomarkers that showed the most significant difference among those analyzed. Additionally, SDF-1 was the marker with the highest number of positive cells in MLs.

Except for vimentin (*p* = 0.282), all biomarkers were more highly expressed in breast lesions with microcalcifications compared to those without calcifications, regardless of the benign or malignant nature of the lesion (Figure 1B).

A different pattern of expression was observed by considering only lesions without microcalcifications (Figure 2A). Remarkably, higher expressions of BMP-2 (0.002) (Figure 2B), BMP-4 (*p* = 0.0082) (Figure 2C), vimentin (*p* = 0.01) (Figure 2D), and SDF-1 (*p* = 0.0008) (Figure 2E) were observed in malignant lesions compared to benign ones (Figure 2A).

To verify whether the studied biomarkers are dependent on the known prognostic factors for breast cancer, their expression was compared with the following prognostic factors: histological grade, TNM classification, lymph node status, and the expression of ER, PR, Ki67, and HER2.

Concerning the histological grade (Figure 3A), OPN was the only marker to display a significant group effect (*p* = 0.032). Moreover, *t*-test analysis revealed that OPN was overexpressed in G2 lesions, as compared to G3 ones (*p* = 0.009). This biomarker is considered to be a molecule involved in microcalcification formation, as reported previously [35].

Further analysis based on pathological stage (pT) (Figure 3B) and lymph node involvement (pN) (Figure 4A) revealed no significant correlations for all investigated biomarkers.

For HER2 score, only BMP-4 exhibited a significant group effect (*p* = 0.018) with a significant increase in the number of BMP-4 positive cells in samples with HER2 score 3, as compared to those with HER2 score 0 (*p* = 0.005) (Figure 4B).

**Figure 4 ijms-26-00645-f004:**
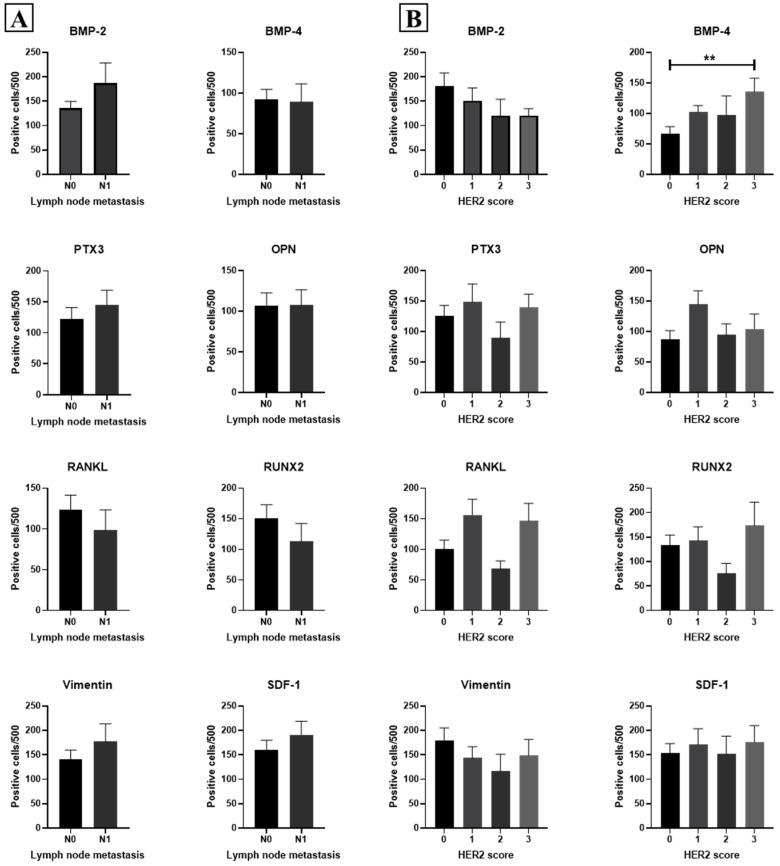
Association among EMT, mineralization biomarkers and both lymph node metastasis and HER2 score. Graphs show the association among EMT, mineralization biomarkers, and lymph node metastasis (**A**), or HER2 score (**B**) **: *p* < 0.01; Correlations between marker expression and hormone receptor (ER, PR) and Ki67 status were also evaluated by a Pearson analysis, but no significant associations were found (Figure 5).

**Figure 5 ijms-26-00645-f005:**
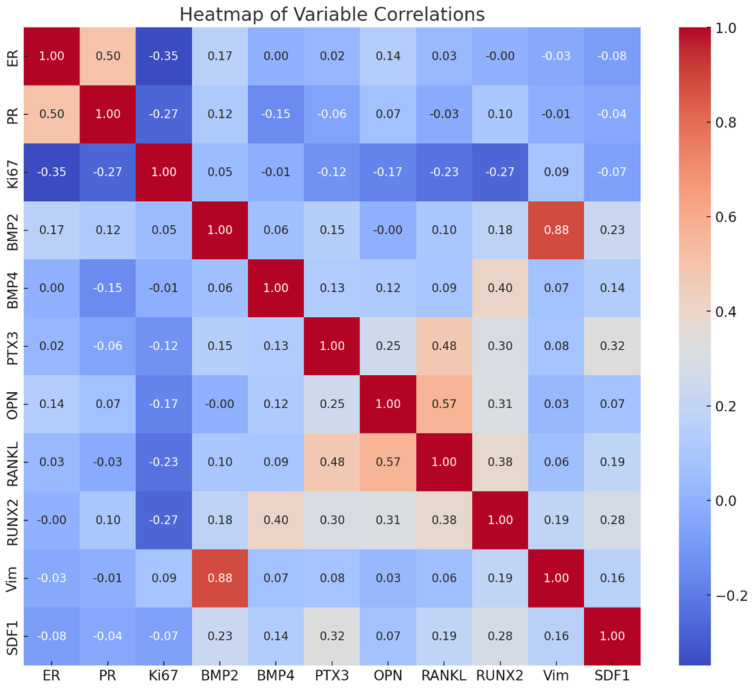
The heatmap shows the correlation between the expression of BMP-2, BMP-4, PTX3, OPN, RANKL, Runx2, vimentin, SDF-1, ER, PR, and Ki67.

### 2.3. Bioinformatic Analysis

To investigate the potential prognostic role of BMP-2, BMP-4, SDF-1, and vimentin, patients’ survival status was evaluated using the cBioPortal database (https://www.cbioportal.org/ (accessed on 10 December 2024)). BMP-2 and BMP-4 expression showed no significant difference between patients who died due to cancer or other causes and those alive, with a maximum follow-up period of 350 months (Figure 6A,B). Of note, high expressions of both SDF-1 and vimentin were observed in the living group, as compared to both those who died of others causes and those who died of the disease (SDF-1 died of diseased vs. living *p* < 0.0001; died of other causes vs. living *p* < 0.0001; Vim died of diseased vs. living *p* < 0.0001; died of other causes vs. living *p* < 0.0001) (Figure 6C,D).

To evaluate the potential prognostic impact of these biomarkers on specific subsets of breast cancer patients, overall survival curves were analyzed both in patients regardless of the molecular subtype of breast cancer and in those with specific subtypes, including basal-like, luminal A, luminal B, HER2, and normal-like carcinomas. Additionally, survival analysis was conducted in patients with or without lymph node metastases at the time of diagnosis. When considering all types of breast carcinomas (Figure 7A), patients with high expression of BMP-4 (*p* = 0.0013), SDF-1 (*p* = 0.038), or vimentin (*p* = 0.0032) demonstrated a significant improvement in overall survival, compared to those with lower expression, although the increase in survival was relatively limited.

In patients with luminal A breast cancer, high expressions of BMP-4, SDF-1, and vimentin demonstrated a more substantial positive impact on overall survival (Figure 7B; BMP-4 *p* = 0.0025; SDF-1 *p* = 0.0019; vimentin *p* = 0.00051). Conversely, in patients affected by luminal B breast carcinomas only a high level of BMP-4 and SDF-1 positively correlated with a better overall survival (Figure 7C; BMP-4 *p* = 0.0045; SDF-1 *p* = 0.012).

Paradoxically, the impact of these biomarkers on the overall survival of HER2-positive, basal-like, and normal-like breast carcinomas differs significantly when compared to luminal-type carcinomas (Figure 8). Concerning the HER2 positive carcinomas, BMP-2 emerged as a negative prognostic factor. In fact, patients with higher BMP-2 expression showed a significant increase in mortality rate mainly after 40 months (Figure 8A; BMP-2 *p* = 0.041). The expression of SDF-1 in HER2 positive carcinomas correlates with a better prognosis for survival (Figure 8A; SDF-1 *p* = 0.027).

In basal-like carcinomas, SDF-1 demonstrated the greatest impact for survival. Specifically, patients with high levels of SDF-1 expression showed a higher probability for survival as early as 20 months (Figure 8B; SDF-1 *p* < 0.0001). It is important to underline that the expression of BMP-2 in basal-like carcinomas showed a very significant impact on prognosis, as compared to HER2 positive cancers. In fact, in basal-like lesions, high levels of BMP-2 expression correlated with a better prognosis (*p* = 0.0065). The overall survival curve for normal-like breast carcinomas (Figure 8C) is not highly informative due to the limited number of patients analyzed.

Bioinformatic analysis highlighted the potential of the investigated biomarkers to predict chemotherapy response (Figure 4). Notably, while BMP-2 (Figure 9A) and BMP-4 (Figure 9B) expression levels remained consistent between responders and non-responders (BMP-2 *p* = 0.57; BMP-4 *p* = 0.61), elevated levels of vimentin (Figure 9C) or SDF-1 (Figure 9D) were strongly associated with improved chemotherapy outcomes (vimentin *p* < 0.0001; SDF-1 *p* < 0.0001).

## 3. Discussion

This study showed that EMT and mineralization biomarkers were significantly overexpressed in breast carcinomas, regardless of lesion type or the presence of calcifications.

Notably, among these, vimentin, BMP-2, BMP-4, and SDF-1 were overexpressed in malignant lesions even in the absence of calcifications and showed no correlation with any of the common prognostic factors typically required in clinical practice.

Interestingly, the increased expression of these biomarkers positively correlated with overall survival, suggesting that they may be considered to be independent positive prognostic factors, beyond the biomarkers currently used in clinical practice.

The presence of microcalcifications in breast lesions is also considered to be a diagnostic and prognostic marker [36]. Previous studies have associated the formation of microcalcifications with the occurrence of the EMT phenomenon, hypothesizing that calcifications made of hydroxyapatite are produced by breast cancer cells that have differentiated into osteoblast-like cells after acquiring mesenchymal characteristics [37,38]. It is crucial to distinguish EMT biomarkers linked to microcalcification production from those independent of this process, as the latter could serve as valuable independent prognostic markers. EMT can occur in various physiological and pathological contexts and act through multiple molecular pathways, which include activation of transcription factors [39], expression of specific cell surface proteins, -reorganization and expression of cytoskeletal proteins, production of enzymes capable of degrading the extracellular matrix [40,41,42], and changes in the expression of specific microRNAs [43,44]. Numerous studies have reported the phenomenon of EMT in breast lesions [45,46].

Among EMT biomarkers, vimentin is one of the most extensively investigated molecules for in situ studies. It is a cytoskeletal protein, and its upregulation is often associated with enhanced cellular motility, increased resistance to apoptosis, and greater metastatic potential in cancer cells [47].

In this study, higher expressions of vimentin were observed in malignant breast lesions, regardless of the presence of microcalcifications. This finding suggests that the upregulation of vimentin may represent an early event in microcalcification formation, while also reflecting a more complex biological network associated with the acquisition of cellular motility properties, enabling breast cancer cells to invade surrounding tissues.

Similar results have been observed for the expression of the mineralization biomarkers BMP-2, BMP-4, and SDF-1. In fact, their expression increased in the malignant lesions regardless of the presence of calcifications. It is important to note that these molecules are also involved in both EMT and the acquisition of stem-like properties, as demonstrated in in vitro investigations.

The transition to a mesenchymal phenotype by carcinoma cells is closely linked to an increased tumor invasiveness and metastatization [48]. On the other hand, EMT has been associated with cancer stem cell properties (CSCs) responsible for tumor initiation and self-renewal capabilities. It is now recognized that the phenomenon of EMT and the presence of CSC are two partly overlapping phenomena, and that the tumor cell population that co-expresses EMT and CSC markers underlies therapy resistance and disease recurrence [49].

SDF1, also called CXC chemokine ligand 12 (CXCL12), is a good example of the overlapping phenomenon of CSC and EMT. In fact, it has been shown that some chemokine, including CXCL12, are able to regulate breast cancer stem cell behavior. In particular, the SDF-1—CXCR4 axis is a key regulator of CSC trafficking [50] and plays a critical role in directing their metastasis to organs with high SDF-1 expression, such as bone. At the same time, the chemokines secreted by breast cancer stem cells, via autocrine and paracrine signaling, promote their EMT [51].

Furthermore, a metanalysis showed a positive prognostic role for SDF1/CXCL12 in breast cancer, with its overexpression correlated with better prognosis in terms of both disease-free survival and overall survival of breast cancer [52].

The BMP family was originally identified for their ability to facilitate bone formation at extra-skeletal sites. In the last decade, BMPs have been studied in several cancers, and many studies have revealed roles for BMPs in EMT promotion and for increasing the migration/invasion of several cancers [53], including breast tumors [54].

The role of BMP-2 and BMP-4 in promoting EMT and regulating the survival and maintenance of CSC has also been reported [55]. Many studies have highlighted the complexity of these BMPs in breast cancer. In this context, one study [16] demonstrated that the in situ expression of BMP-2 in human breast cancer is closely associated with both EMT and microcalcification formation, depending on its cellular localization. Specifically, their findings suggest that cytoplasmic BMP-2 contributes to the EMT process, whereas its nuclear form is associated with the expression of mineralization markers and the formation of microcalcifications. The complexity of BMP-2 signaling is further emphasized by its ability to mediate CSC formation, highlighting its potentially contradictory role in cancer progression [56].

This evidence suggests an intricate, complex, and often paradoxical role for BMPs and SDF-1 in the development and progression of cancer.

The bioinformatics analysis provided valuable insights into the possible prognostic and therapeutic significance of BMP-2, BMP-4, SDF-1, and vimentin in breast cancer. While BMP-2 and BMP-4 expression did not appear to influence patient survival, SDF-1 and vimentin levels were notably higher in the surviving group, underscoring their potential as favorable prognostic markers. Notably, the analysis of overall survival curves also revealed significant subtype-specific differences. In particular, SDF-1 has a significant impact on overall survival in breast cancer patients with HER2-positive or basal-like carcinomas, two molecular subtypes typically associated with a poor prognosis. The evidence that SDF-1 levels are significantly associated with better survival outcomes in these subtypes suggests that it may modulate tumor progression in a subtype-specific manner. The role of SDF-1 in promoting tumor cell migration and invasion through its interaction with the CXCR4 receptor is well-documented [57]. However, its association with improved survival in these aggressive subtypes might indicate a more complex interplay, possibly involving the recruitment of immune cells [58], the modulation of angiogenesis [59], or interference with the pathways specific to HER2-positive or basal-like tumors. These findings also raise questions about the potential for therapies targeting the SDF-1—CXCR4 axis to provide subtype-specific benefits. In fact, several CXCL12—CXCR4 antagonists have been developed, which have shown encouraging results in anti-cancer activity both in in vitro and in vivo studies [60].

The importance of the molecular characterization of breast cancers is underscored by the paradoxical prognostic impact of BMP-2 and vimentin across different cancer subtypes. These molecules demonstrate either positive or negative associations with overall patient survival, depending on the specific type of molecular cancer. The expression of BMP-2, in particular, is associated with a better prognosis in HER2-positive breast cancers. However, patients with high levels of BMP-2 expression exhibit higher mortality rates when affected by basal-like lesions. The different and apparent contrasting prognostic roles of these molecules highlight the importance of molecular heterogeneity in breast cancer. A molecule that positively influences patient survival in one molecular subtype may have a detrimental impact on prognosis in another. This variability depends on the unique molecular pathways, receptor profiles, and interactions within the tumor microenvironment that characterize different breast cancer subtypes. Therapies targeting BMP-2 or vimentin have recently been investigated. Regulating and targeting BMP signaling at the level of extracellular receptors presents significant clinical potential [61,62]. Several natural antagonists and chemical inhibitors have already demonstrated their effectiveness in suppressing metastasis in cancers, such as prostate [63], lung [64], and breast [65]. Despite notable advancements in developing highly specific and less toxic small-molecule inhibitors targeting BMP type I receptors, these inhibitors remain nonselective, raising concerns about their clinical applicability. The importance of vimentin in biological processes involved in cell motility makes it an attractive target for personalized therapies. However, unlike in the case of actin or the microtubule cytoskeleton, there is currently no routinely used drug to specifically target intermediate filaments. This makes their study, as well as their use as a target in clinical trials, much more challenging. In this context, the most extensively studied drug targeting vimentin intermediate filaments is withaferin A (WFA), a tumor inhibitor derived from the plant *Withania somnifera* [66,67]. WFA downregulates vimentin expression and induces the disassembly of vimentin filaments in a dose-dependent manner [68], further highlighting its potential as a therapeutic agent. It is demonstrated that WFA directly binds vimentin. In addition to its ability to inhibit tumor growth, WFA also exhibits pro-apoptotic properties [69].

The experimental evidence suggest that therapies targeting BMP-2 and vimentin could be integrated into combination treatment regimens, alongside conventional chemotherapy or immunotherapy, to improve outcomes. For instance, inhibiting BMP-2 or vimentin could sensitize cancer cells to existing therapies or prevent metastatic spread. Preclinical studies and early-phase clinical trials are needed to evaluate the safety, efficacy, and optimal use of such therapies.

The expression of vimentin and SDF-1 is also particularly notable for its potential to predict chemotherapy response. Patients with higher levels of these molecules have shown improved treatment outcomes, highlighting their predictive value. Assessing SDF-1 expression, either in tissue samples or peripheral blood, could represent a reliable test to identify patients most likely to benefit from chemotherapy. Tissue-based assessments provide a localized understanding of SDF-1’s role within the tumor microenvironment, particularly its interactions with the CXCR4 receptor, which influences cancer stem cell behavior, migration, and metastasis [57]. These insights are critical for tailoring therapeutic strategies to the molecular profile of individual tumors [70].

On the other hand, evaluating SDF-1 levels in peripheral blood offers a less invasive and more accessible method for patient stratification. A blood-based test could allow for real-time monitoring of SDF-1 levels throughout treatment, enabling dynamic adjustments to therapeutic protocols.

Nevertheless, further studies are needed to elucidate the precise regulatory mechanisms underlying the dual role of these molecules in breast cancer.

### Study Limitations

This study has some limitations. The retrospective design may introduce potential bias or confounding factors, which could affect the applicability of our findings until validated in a prospective study. The study cohort was derived from a single institution, which may limit the broader applicability of the results to more diverse populations. Multicenter studies will be necessary to validate these findings across different demographic and clinical settings. Another limitation is the heterogeneity of breast cancer subtypes within the cohort, which presents challenges in interpreting the findings. Future studies should address this issue by analyzing larger, subtype-specific cohorts.

## 4. Materials and Methods

### 4.1. Breast Sample Collection

For this study we retrospectively collected 133 consecutive paraffin embedded diagnostic blocks from breast biopsies. From each paraffin-embedded block, serial sections were obtained to perform both histological classification and immunohistochemical analysis. Thee study protocol was approved by the “Policlinico Tor Vergata” Independent Ethical Committee (reference number #96-19).

### 4.2. Histology and Immunohistochemistry

After fixation in 10% buffered formalin for 24 h, breast tissues were embedded in paraffin. Three-micrometer thick sections were stained with hematoxylin and eosin (H&E) and the diagnostic classification was blindly performed by two pathologists (EG and FS).

The main prognostic and predictive factors were documented for the infiltrating breast carcinomas: histological differentiation grade, lesion size (classified as pT), lymph node involvement (classified as pN), expression levels of hormone receptors (ER and PR), HER2 expression, and proliferation index (Ki67).

Immunohistochemical analysis was performed to assess the expression of the markers: runt-related transcription factor 2 (RUNX2), bone morphogenetic protein 2 (BMP-2), bone morphogenetic protein 4 (BMP-4), receptor activatorra of nuclear factor kappa beta (RANKL), osteopontin (OPN), pentraxin 3 (PTX3), vimentin (VIM), and SDF-1.

Briefly, antigen retrieval was performed on 3-μm-thick paraffin sections using EDTA citrate pH 7.8 or citratec pH 6.0 buffers for 30 min at 95 °C. Sections were then incubated for 1 h at room temperature with the antibodies reported in Table 1. Washings were performed with PBS/Tween20 pH 7.6. Reactions were revealed by the HRP//DAB Detection Kit (UCS Diagnostic, Rome, Italy).

We evaluated the immunohistochemical signal by counting the number of positive breast cells from a total of 500 cells in randomly selected regions by using Axioscope 5 light microscope (Zeiss, Oberkochen, Germany). The regions were blindly selected on H&E-stained sections, thus avoiding any influence from the immunohistochemical results.

### 4.3. Bioinformatic Analysis

To investigate the potential prognostic value of BMP-2, BMP-4, SDF-1, and vimentin, the patient’s survival status were extracted from the cBioPortal database (2509 patients affected by infiltrating breast carcinomas) (https://www.cbioportal.org/ (accessed on 10 December 2024)). Overall survival curves were analyzed by using RNASeq data from the Kaplan—Meier Plotter (https://kmplot.com/analysis/ (accessed on 10 December 2024)) [71]. The Kaplan–Meier Dataset included 2976 breast cancer patients with a follow-up of 80 months. Starting from this data set, the overall survival analysis was performed by subdividing the patients in the following prognostic groups: basal like, luminal A, luminal B, HER2 positive, normal, presence of lymph nodes metastasis, and absence of lymph nodes metastasis.

### 4.4. Statistical Analysis

Statistical analysis was performed to evaluate the distribution and differences of the immunohistochemical data. The Kruskal—Wallis’s test was applied to compare three or more independent groups. To identify significant differences in marker expression across categorical variables, pairwise comparisons between groups were performed using the Mann—Whitney U test. A Pearson analysis was used to investigate the association between continuous variables. Statistical significance was set at *p* < 0.05.

## 5. Conclusions

In the era of personalized medicine, the identification of reliable prognostic biomarkers in breast cancer has become a critical focus in oncology research. While the introduction of molecular classifications for breast cancer has marked a significant advancement in the management of breast cancer patients, it remains evident that some patients fail to effectively respond to targeted biological therapies. This highlights the need for more precise and comprehensive tools to guide treatment decisions and improve outcomes.

The findings here reported underscore the need for further studies to elucidate the mechanisms underlying the roles of these biomarkers in breast cancer. Preclinical and early-phase clinical trials are essential to evaluate the safety, efficacy, and optimal integration of BMPs, SDF-1, and vimentin in the clinical setting of breast cancer. By bridging molecular insights with practical applications, these biomarkers hold the potential to improve breast cancer diagnostics and therapeutics, offering new avenues for personalized medicine.

## Figures and Tables

**Figure 1 ijms-26-00645-f001:**
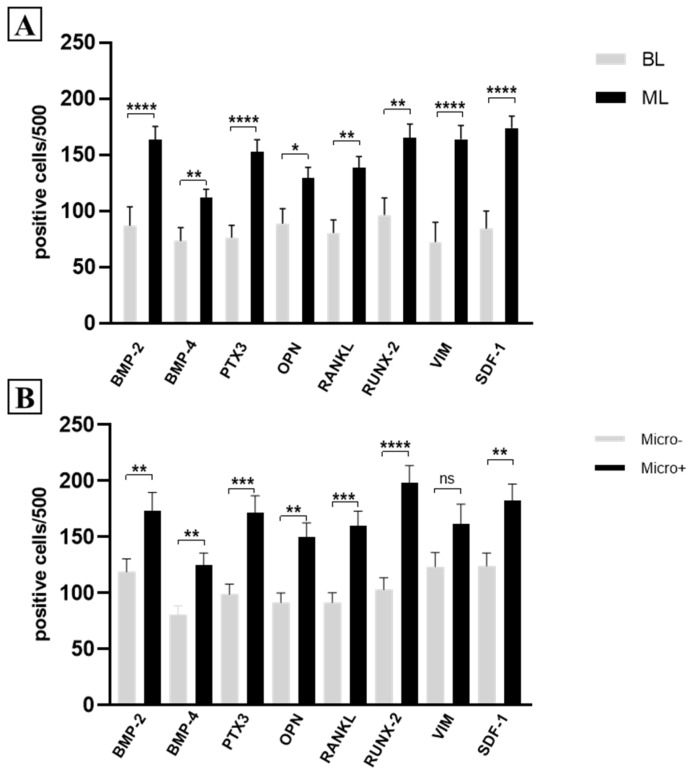
Expression of EMT and mineralization biomarkers. (**A**) Graph shows the expression of EMT and mineralization biomarkers in benign breast lesions (BL), as compared to malignant ones (ML). (**B**) Graph displays the expression of EMT and mineralization biomarkers in breast cancer lesions with microcalcifications (Micro+), as compared to lesions without microcalcifications (Micro−). *: *p* < 0.05; **: *p* < 0.01; ***: *p* < 0.001; ****: *p* < 0.0001; ns: Not significant (*p* ≥ 0.05).

**Figure 2 ijms-26-00645-f002:**
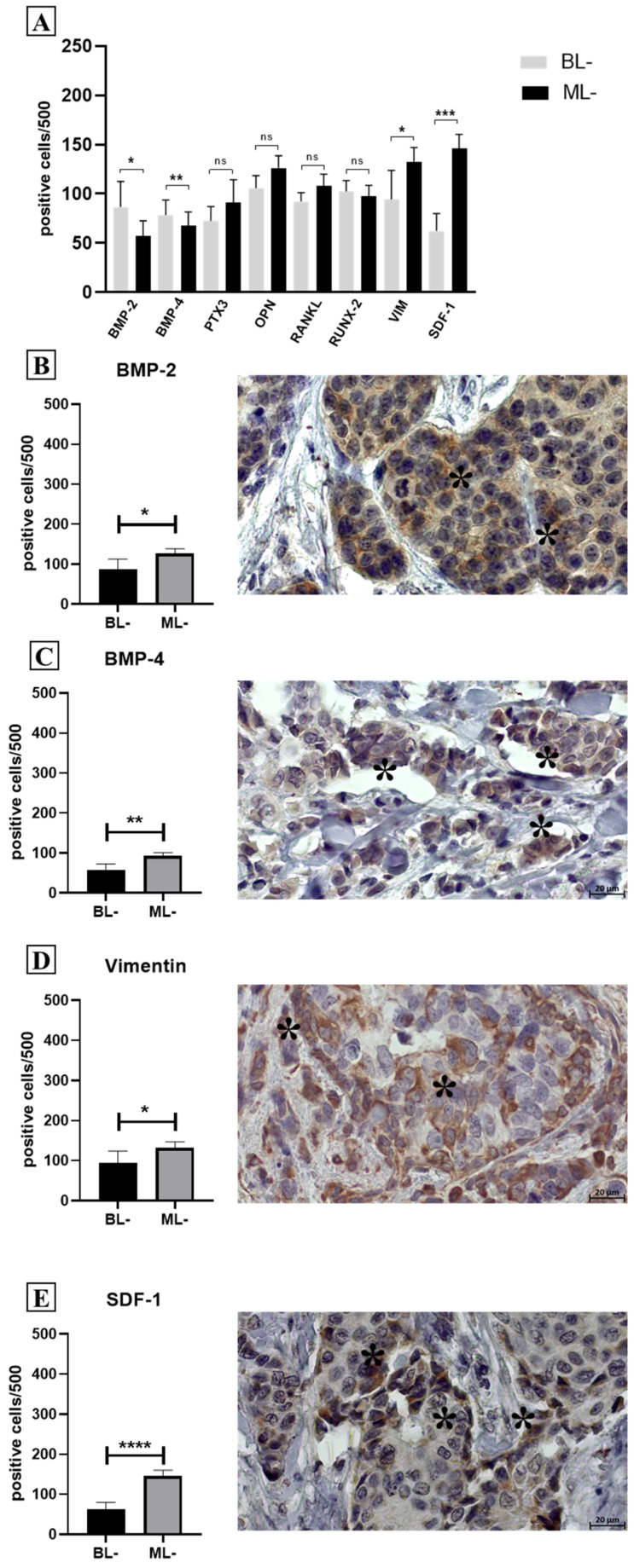
Expression of EMT and mineralization biomarkers in malignant lesions compared to benign ones, regardless of the presence of microcalcifications. (**A**) Graph displays the expression of EMT and mineralization biomarkers in benign breast lesion without microcalcification (BL−), as compared to malignant lesions without microcalcifications (ML−). (**B**–**E**) Graphs show higher expressions of BMP-2 (**B**), BMP-4 (**C**), vimentin (**D**), and SDF-1 (**E**) in cancer without calcification, as compared with benign lesion. Representative images of BMP-2 (**B**), BMP-4 (**C**), vimentin (**D**), and SDF-1 (**E**) in cancer-positive cells in a breast-infiltrating carcinoma. *: *p* < 0.05 **: *p* < 0.01; ***: *p* < 0.001; ****: *p* < 0.0001; ns: Not significant (NS) (*p* ≥ 0.05). Asterisks mark the positive cells in the panels (**B**–**E**).

**Figure 3 ijms-26-00645-f003:**
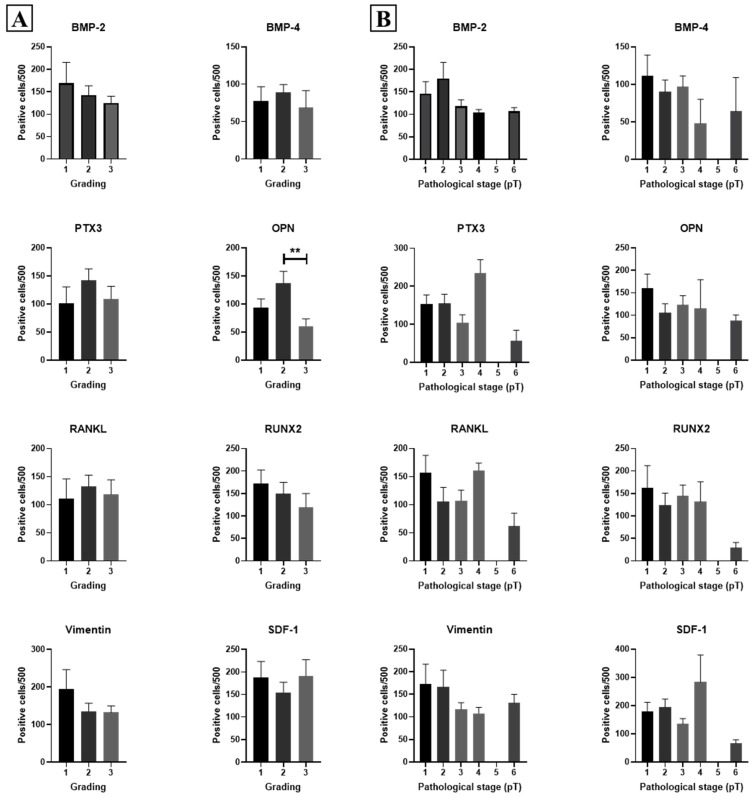
Association among EMT, mineralization biomarkers, and both histological grading and pathological stage. Graphs show the association among EMT, mineralization biomarkers, and histological grading (grading) (**A**), or pathological stage (**B**); **: *p* < 0.01.

**Figure 6 ijms-26-00645-f006:**
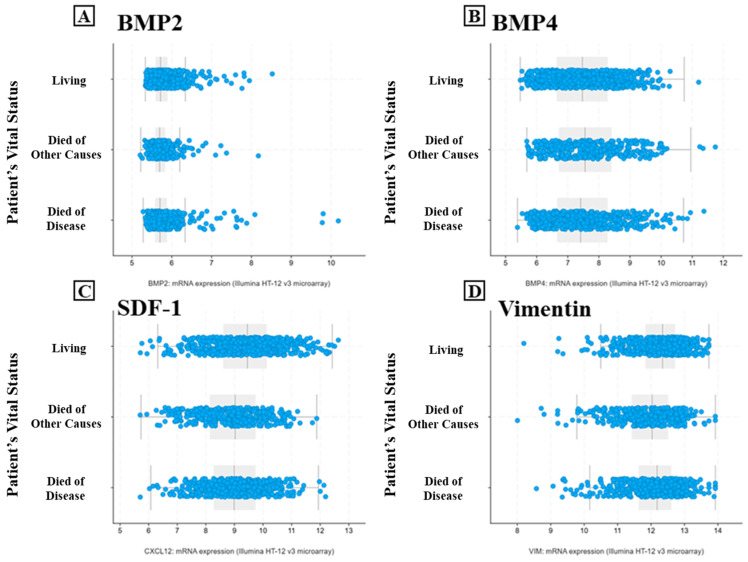
Patients’ survival status according to the expression of BMP-2, BMP-4, SDF-1, and vimentin. Graphs display no significant difference in the expression of BMP-2 (**A**) and BMP-4 (**B**) among living, died of other causes, and died of disease groups. Graphs show a significant increase in SDF-1 (**C**) and vimentin (**D**) expression in the living group, as compared to both died of other causes and died of disease groups. Blue Dots: Represents individual data points for mRNA expression levels of the respective genes. The boxes indicate the 1st and 3rd quartiles, and the grey lines indicate the median.

**Figure 7 ijms-26-00645-f007:**
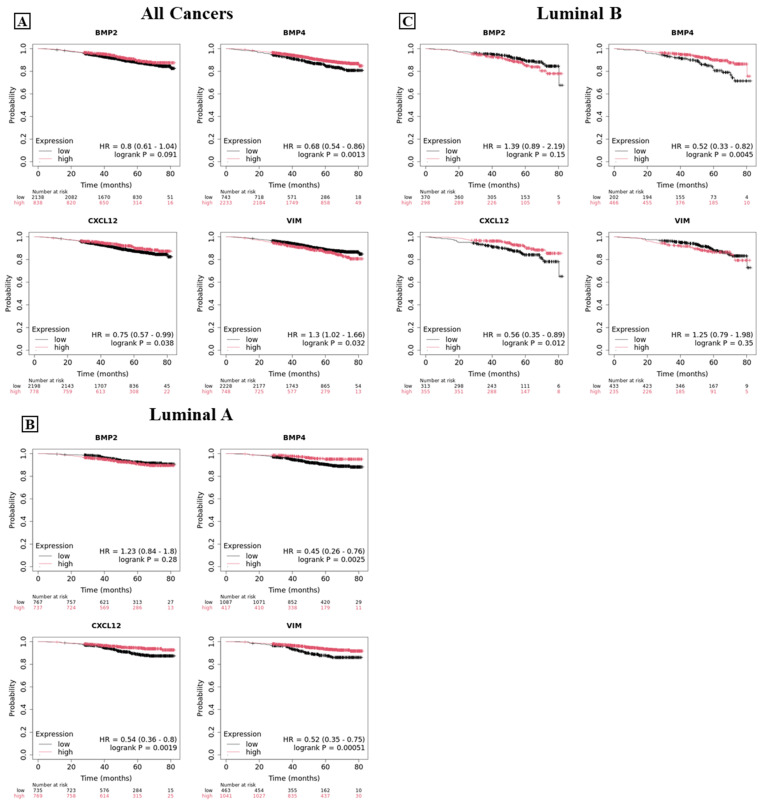
Overall survival curves for BMP-2, BMP-4, SDF-1 (CXCL12 gene), and vimentin. Graphs show the overall survival curve for BMP-2, BMP-4, SDF-1 (CXCL12 gene), and vimentin in breast cancer patients regardless of the molecular subtype (**A**),in luminal A (**B**), and in luminal B (**C**) breast cancer patients.

**Figure 8 ijms-26-00645-f008:**
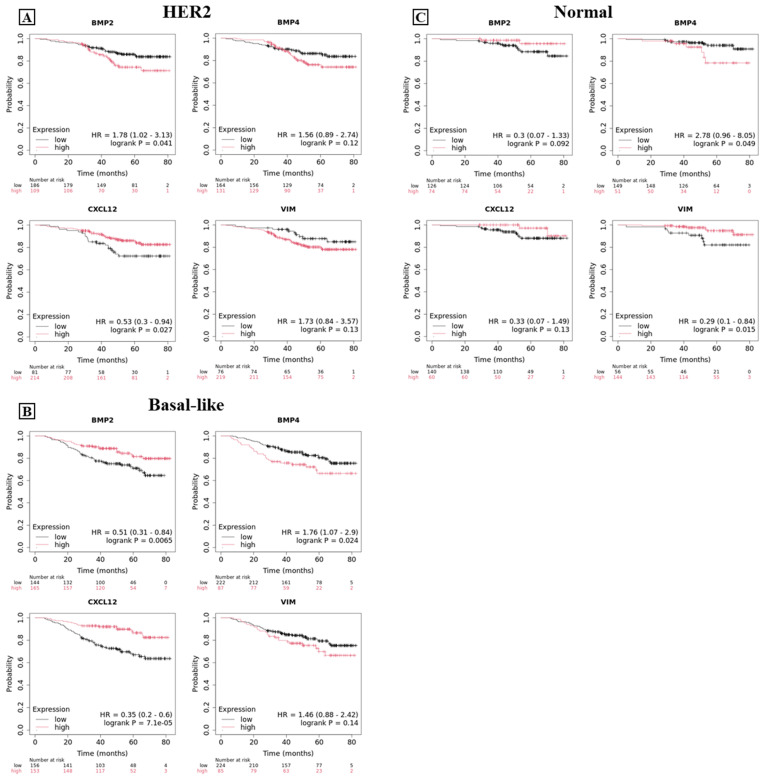
Overall survival curves for BMP-2, BMP-4, SDF-1 (CXCL12 gene), and vimentin. Graphs show the overall survival curve for BMP-2, BMP-4, SDF-1 (CXCL12 gene), and vimentin in patients affected by HER2 positive breast cancer (**A**), in basal-like (**B**), and in normal-like (**C**) breast cancer patients.

**Figure 9 ijms-26-00645-f009:**
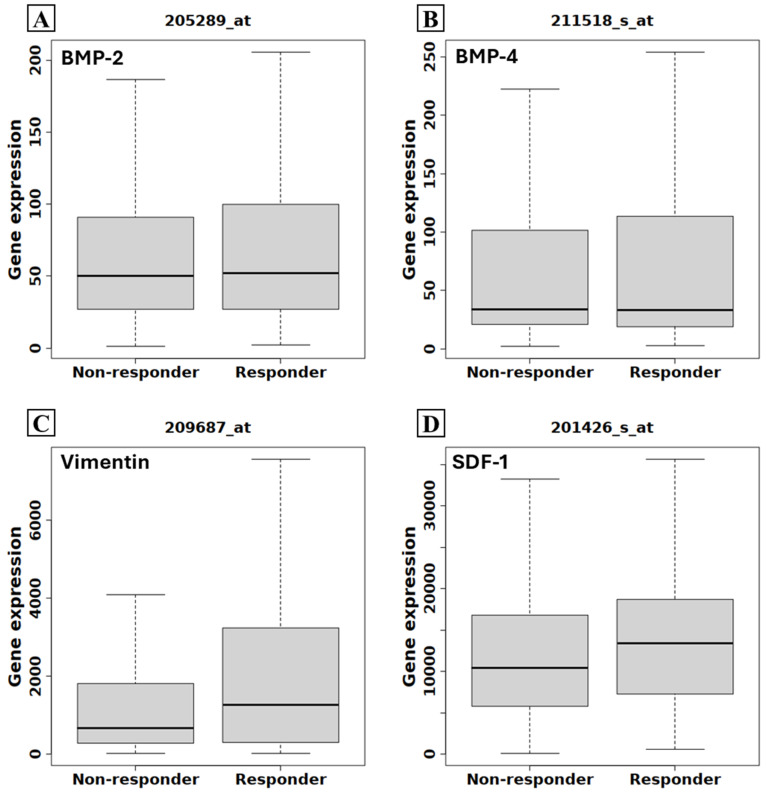
Association between BMP-2, BMP-4, vimentinv, and SDF-1 and the response to chemotherapy based on RNASeq data. Graphs show no significant difference between BMP-2 (**A**) and BMP-4 (**B**) expression and the response to chemotherapy. Breast cancer patients with higher vimentin (**C**) and SDF-1 (**D**) expression show a better response to chemotherapy. The boxes indicate the 1st and 3rd quartiles, and the grey lines indicate the median.

**Table 1 ijms-26-00645-t001:** List of primary antibodies.

Antibody	Characteristics	Dilution	Retrieval
anti-BMP-2	rabbit clone N/A; Novus Biologicals, Littleton, CO, USA	1:500	Citrate pH 6.0
anti-BMP-4	rabbit polyclonal clone 3C11C7; Novus Biologicals, Littleton, CO, USA	1:100	Citrate pH 6.0
anti-OPN	Mouse monoclonal clone AE1/AE3/PCK26; Ventana, Tucson, AZ, USA	1:100	EDTA citrate pH 7.8
anti-PTX3	rat monoclonal clone MNB1; AbcamAbcam, Cambridge, UK	1:100	Citrate pH 6.0
anti-RANKL	rabbit monoclonal clone 12A668; Abcamc, Cambridge, UK	1:100	Citrate pH 6.0
Anti-RUNX2	Mouse monoclonal clone 3F5; Novus Biologicals, Littleton, CO, USA	1:100	Citrate pH 6.0
Anti-SDF-1	Mouse monoclonal clone 79018; Novus Biologicals, Littleton, CO, USA	1:100	EDTA citrate pH 7.8
anti-Vimentin	mouse monoclonal clone V9; Ventana, Tucson, AZ, USA	Pre-diluted	EDTA citrate pH 7.8

## Data Availability

The original contributions presented in this study are included in the article. Further inquiries can be directed to the corresponding authors.

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
