# Peer review of "Implications of Mineralization Biomarkers in Breast Cancer Outcomes Beyond Calcifications"

_ijms, 2025, doi:10.3390/ijms26020645_

Round 1

Reviewer 1 Report

Comments and Suggestions for Authors

This study “Implications of Mineralization Biomarkers in Breast Cancer Outcomes Beyond Calcifications” indicated the status of mineralization biomarkers in breast cancer and concluded the potential in prognosis. Some points need to be addressed.

1. The Introduction and Discussion sections of this article are written in an informal format, consisting only of sentences without proper paragraphs, and need to be reorganized to a scientific format.

2. Some information about breast cancer research discussed in the Discussion section should be moved to the Introduction, such as lines 238–264 and 282–305.

3. Regarding “the evaluation of the immunohistochemical signal by counting the number of positive breast cells out of a total of 500 cells in randomly selected regions”, line 312-313, how were the regions randomly selected, and what instruments and software were used?

4. line 131-136, “OPN was the only marker to display a significant group effect (p=0.032). Moreover, t-test analysis revealed that OPN was overexpressed in G2 lesions as compared to G3 ones (p=0.009). Further analysis based on pathological stage (pT) (Figure 3B) and lymph node in volvement (pN) (Figure 4A) revealed no significant correlations for all investigated biomarkers.” Concerning the role of OPN, it should be performed immunohistochemistry and discussed more

5. The figure captions must include explanations of the meanings of these symbols, *, **, ***, ****, ns etc.

6. In Figure 2, the histological results, symbols must be used to mark the positive areas. The immunohistochemical results measured in Figure 2 should be included BL and ML...

7. in Figure 4B, the gene expression of BMP2 is inversely proportional to Her2. The possibilities should be discussed.

Comments on the Quality of English Language

Please revise the text of English into a scientific format.

Author Response

Manuscript ID ijms-3412935

" Implications of Mineralization Biomarkers in Breast Cancer Outcomes Beyond Calcifications"

Submitted to: International Journal of Molecular Scinece

ROUND#2

POINT-TO-POINT REBUTTAL TO REVIEWER COMMENTS

General Comments to Editor and Reviewers

We appreciated the thoughtful and constructive criticisms and suggestions of Reviewers. His/her comments on how to improve the manuscript, which has been revised accordingly. We also appreciate the Editors for calling for a new re-submission of an improved version of our manuscript.

REVIEWER#1

This study “Implications of Mineralization Biomarkers in Breast Cancer Outcomes Beyond Calcifications” indicated the status of mineralization biomarkers in breast cancer and concluded the potential in prognosis.

Reply: We sincerely thank the reviewer for their valuable suggestions, which have significantly improved the quality of our manuscript. We appreciate your thoughtful feedback on our study.

  1. The Introduction and Discussion sections of this article are written in an informal format, consisting only of sentences without proper paragraphs, and need to be reorganized to a scientific format.

Reply: We have significantly revised the introduction and discussion paragraph. The updated version of the introduction now focuses more directly on the central issue, emphasizing the importance of EMT and the biomarkers BMP-2, BMP-4, SDF-1, and Vimentin. Additionally, the purpose of the study has been clearly stated.

The discussion paragraph has been substantially revised based on the reviewer’s suggestions. In the updated version of our manuscript, we provide a more concise and focused discussion, with greater emphasis on the potential clinical applications of our findings.

  1. Some information about breast cancer research discussed in the Discussion section should be moved to the Introduction, such as lines 238–264 and 282–305.

Reply: Done

  1. Regarding “the evaluation of the immunohistochemical signal by counting the number of positive breast cells out of a total of 500 cells in randomly selected regions”, line 312-313, how were the regions randomly selected, and what instruments and software were used?

Reply: Done. The Section was modified as follow:

We evaluated the immunohistochemical signal by counting the number of positive breast cells out of a total of 500 cells in randomly selected regions by using Axioscope 5 light microscope (Zeiss, Oberkochen, Germany). The regions were blindly selected on H&E-stained sections, thus avoiding any influence from the IHC results.

  1. line 131-136, “OPN was the only marker to display a significant group effect (p=0.032). Moreover, t-test analysis revealed that OPN was overexpressed in G2 lesions as compared to G3 ones (p=0.009). Further analysis based on pathological stage (pT) (Figure 3B) and lymph node in volvement (pN) (Figure 4A) revealed no significant correlations for all investigated biomarkers.” Concerning the role of OPN, it should be performed immunohistochemistry and discussed more

Reply: The result indicated in lines 131-136 is already based on the IHC analysis of OPN.

We did not further discuss the role of OPN because it is a marker associated with microcalcifications and is therefore not independent of them. Our study focuses on markers associated with malignant lesions that do not feature calcifications. The association between OPN and microcalcifications has already been evaluated and commented on in another publication (PMID: 24758513). According to this, in the paragraph “2.3. Correlation with Histological Prognostic Factors” of the result we added the following sentence:

This biomarker is considered as molecule involved in microcalcification formation as reported previously

  1. The figure captions must include explanations of the meanings of these symbols, *, **, ***, ****, ns etc.

Reply: Thank you for this point out. We will revise the figure captions to include explanations for the symbols (*, **, ***, ****, ns, etc.) to ensure clarity regarding their meanings and statistical significance levels.

*: p<0.05p; **: p<0.01; ***: p<0.001; ****: p<0.0001; ns: Not significant (p≥0.05).

  1. In Figure 2, the histological results, symbols must be used to mark the positive areas. The immunohistochemical results measured in Figure 2 should be included BL and ML...

Reply: In the new version of our manuscript, we marked the positive cells with asterisks. The IHC panels provided are representative images of malignant lesions. Due to the large number of comparisons conducted in our study, it is challenging to represent all subgroups.

  1. in Figure 4B, the gene expression of BMP2 is inversely proportional to Her2. The possibilities should be discussed.

Reply: Thank you for pointing this out. Although a trend is shown in the graph, no significant differences were observed among the groups.

Reviewer 2 Report

Comments and Suggestions for Authors

The manuscript presented for review is an interesting analysis of novel biomarkers linked to microcalcifications as indipendent prognostic factors for breast cancer patients. 

The work behind this research is very rigorous and scientifically correct.  My suggestions for improvement are more of form rather than content and I congratulate the authors for my inability to find major flaws with the manuscript.

I suggest the following modifications

1. First, please move the methodology before the results. 

2. In figure 2 there is a part of the legend in Italian. Please change it to English.

3. All figure legends include repetitive expression.  Please use a general description and the just name the biomarker analyzed in each subfigure.

4. Adjust the histogram in figure 2 - they are very small compared to the IHC images

5. Figure 8 may be broken down in 3 figures. As it is the numbers are quite small and difficult to read

6. Minor english proofing 

Author Response

Manuscript ID ijms-3412935

" Implications of Mineralization Biomarkers in Breast Cancer Outcomes Beyond Calcifications"

Submitted to: International Journal of Molecular Scinece

ROUND#1

POINT-TO-POINT REBUTTAL TO REVIEWER COMMENTS

General Comments to Editor and Reviewers

We appreciated the thoughtful and constructive criticisms and suggestions of Reviewers. His/her comments on how to improve the manuscript, which has been revised accordingly. We also appreciate the Editors for calling for a new re-submission of an improved version of our manuscript.

REVIEWER#1

The manuscript presented for review is an interesting analysis of novel biomarkers linked to microcalcifications as indipendent prognostic factors for breast cancer patients.

The work behind this research is very rigorous and scientifically correct.  My suggestions for improvement are more of form rather than content and I congratulate the authors for my inability to find major flaws with the manuscript.

Reply: We sincerely thank the reviewer for their valuable suggestions, which have significantly improved the quality of our manuscript. We appreciate your thoughtful feedback on our study.

First, please move the methodology before the results.

Reply: Thank you for pointing this out. The guidelines of IJMS indicate that the "Methods" section should be placed after the "Discussion" section.

  1. In figure 2 there is a part of the legend in Italian. Please change it to English.

Reply: Done

  1. All figure legends include repetitive expression. Please use a general description and the just name the biomarker analyzed in each subfigure.

Reply: In the new version of our manuscript, all figure legends have been modified in accordance with reviewer suggestion.

  1. Adjust the histogram in figure 2 - they are very small compared to the IHC images

Reply: Done

  1. Figure 8 may be broken down in 3 figures. As it is the numbers are quite small and difficult to read

Reply: We have tried enlarging the image to make it more readable; otherwise, we would have needed to generate four additional figures for Figures 7 and 8. Thank you for your understanding.

  1. Minor english proofing

Reply: Thank you for pointing this out. We have carefully reviewed the manuscript for minor English corrections and made the necessary adjustments.

Reviewer 3 Report

Comments and Suggestions for Authors

The article explores the importance of biomarkers related to epithelial-mesenchymal transition (EMT) and the calcification process in breast cancer. Through the analysis of 133 samples and bioinformatic data, the markers Vimentin, BMP-2, BMP-4 and SDF-1 emerged as potentially independent prognostic factors. The markers were related with overall survival and response to chemotherapy, independent of traditional prognostic markers like histological grade or hormonal status. Among those expression levels of SDF-1 and Vimentin: a better prognosis in certain molecular subtypes of breast cancer can be drawn from that, which can convert them into new potential targets for therapy.

The article, therefore, is in my opinion very important as it presents new possibilities of markers which lie beyond the convention, so as to make the diagnosis and prognosis more accurate. With use of those markers in application of personal approaches in treatments.

I believe that the article ought to be published; however, there are some major issues to address:

1. The introduction contains general information that does not directly focus on the central issue. The purpose of the study should be clearly stated. The introduction should be strengthened with more focus on the importance of EMT and the biomarkers BMP-2, BMP-4, SDF-1, and Vimentin. Also, the novelty of the study needs to be clarified, e.g., how the biomarkers outperform traditional prognostic factors.

2. The presentation of the main results should be done with more conciseness and emphasis on clinical applications.

3. The discussion presents information without a clear connection to the results of the study. The findings should be linked to practical examples or studies. E.g., how high expression of SDF-1 affects specific treatments. It is also necessary to mention possible clinical applications of biomarkers (e.g., their use in non-invasive diagnosis or adaptation of therapeutic protocols).

4. The conclusions need to be strengthened with specific proposals for the integration of the findings into clinical practice. It is also necessary to emphasize the need for further research and the perspectives opened by the study.

5. Some graphs and diagrams are not sufficiently readable, while their captions are less explanatory.

Author Response

Manuscript ID ijms-3412935

" Implications of Mineralization Biomarkers in Breast Cancer Outcomes Beyond Calcifications"

Submitted to: International Journal of Molecular Scinece

ROUND#1

POINT-TO-POINT REBUTTAL TO REVIEWER COMMENTS

 General Comments to Editor and Reviewers

We appreciated the thoughtful and constructive criticisms and suggestions of Reviewers. His/her comments on how to improve the manuscript, which has been revised accordingly. We also appreciate the Editors for calling for a new re-submission of an improved version of our manuscript.

REVIEWER#2

The article explores the importance of biomarkers related to epithelial-mesenchymal transition (EMT) and the calcification process in breast cancer. Through the analysis of 133 samples and bioinformatic data, the markers Vimentin, BMP-2, BMP-4 and SDF-1 emerged as potentially independent prognostic factors. The markers were related with overall survival and response to chemotherapy, independent of traditional prognostic markers like histological grade or hormonal status. Among those expression levels of SDF-1 and Vimentin: a better prognosis in certain molecular subtypes of breast cancer can be drawn from that, which can convert them into new potential targets for therapy.

The article, therefore, is in my opinion very important as it presents new possibilities of markers which lie beyond the convention, so as to make the diagnosis and prognosis more accurate. With use of those markers in application of personal approaches in treatments.

I believe that the article ought to be published; however, there are some major issues to address

Reply: We sincerely thank the reviewer for their valuable suggestions, which have significantly improved the quality of our manuscript. We appreciate your thoughtful feedback on our study.

  1. The introduction contains general information that does not directly focus on the central issue. The purpose of the study should be clearly stated. The introduction should be strengthened with more focus on the importance of EMT and the biomarkers BMP-2, BMP-4, SDF-1, and Vimentin. Also, the novelty of the study needs to be clarified, e.g., how the biomarkers outperform traditional prognostic factors.

Reply:  We have significantly revised the introduction in accordance with your suggestions. The updated version now focuses more directly on the central issue, emphasizing the importance of EMT and the biomarkers BMP-2, BMP-4, SDF-1, and Vimentin. Additionally, the purpose of the study has been clearly stated.

  1. The presentation of the main results should be done with more conciseness and emphasis on clinical applications.

  1. The discussion presents information without a clear connection to the results of the study. The findings should be linked to practical examples or studies. E.g., how high expression of SDF-1 affects specific treatments. It is also necessary to mention possible clinical applications of biomarkers (e.g., their use in non-invasive diagnosis or adaptation of therapeutic protocols).

Reply point 2 and 3: The discussion paragraph has been substantially revised based on the reviewer’s suggestions. In the updated version of our manuscript, we provide a more concise and focused discussion, with greater emphasis on the potential clinical applications of our findings.

  1. The conclusions need to be strengthened with specific proposals for the integration of the findings into clinical practice. It is also necessary to emphasize the need for further research and the perspectives opened by the study.

Reply: We have revised the conclusion paragraph in line with the reviewer's suggestion.

  1. Some graphs and diagrams are not sufficiently readable, while their captions are less explanatory.

Reply: We have tried enlarging the image to make it more readable.

Reviewer 4 Report

Comments and Suggestions for Authors

The article titled "Implications of Mineralization Biomarkers in Breast Cancer Outcomes Beyond Calcifications" by Rui Dong et al. offers valuable insights; however, there are several areas that require attention:

  1. The retrospective study may introduce bias or confounding factors that could impact the validity of the results.
  2. In Figures 3 and 4, authors could conduct statistical analyses and compare among the groups to indicate statistical significance.
  3. Authors should provide potential limitations of the study, including sample size, the heterogeneity of breast cancer subtypes, etc.
  4. Authors may consider conducting further validation studies to assess the prognostic value of the identified biomarkers.
  5. Authors could provide more data on treatment regimens and follow-up information, which could help in interpreting the prognostic significance of the biomarkers.
  6. Authors should discuss the underlying mechanisms related to biomarker expression and chemotherapy response to gain insights into the clinical relevance of the results.

Author Response

Manuscript ID ijms-3412935

" Implications of Mineralization Biomarkers in Breast Cancer Outcomes Beyond Calcifications"

Submitted to: International Journal of Molecular Scinece

ROUND#1

POINT-TO-POINT REBUTTAL TO REVIEWER COMMENTS

 General Comments to Editor and Reviewers

We appreciated the thoughtful and constructive criticisms and suggestions of Reviewers. His/her comments on how to improve the manuscript, which has been revised accordingly. We also appreciate the Editors for calling for a new re-submission of an improved version of our manuscript.

REVIEWER#3

The article titled "Implications of Mineralization Biomarkers in Breast Cancer Outcomes Beyond Calcifications" by Rui Dong et al. offers valuable insights; however, there are several areas that require attention:

Reply: We sincerely thank the reviewer for their valuable suggestions, which have significantly improved the quality of our manuscript. We appreciate your thoughtful feedback on our study.

  1. The retrospective study may introduce bias or confounding factors that could impact the validity of the results.

  1. Authors should provide potential limitations of the study, including sample size, the heterogeneity of breast cancer subtypes, etc.

Reply point 1 and 2: Thank you for pointing this out. In the revised version of our manuscript, we have included a Study Limitations paragraph where we address all the potential limitations of our study.

In Figures 3 and 4, authors could conduct statistical analyses and compare among the groups to indicate statistical significance.

Reply: We appreciate the reviewer's suggestion. In our analysis, t-tests were conducted only for comparisons between groups where the ANOVA results were statistically significant (p < 0.05). This approach ensures that additional pairwise comparisons are justified and avoids inflating the risk of type I errors. The results of these post-hoc analyses have been incorporated into Figures 3 and 4 to indicate statistical significance where applicable.

Authors may consider conducting further validation studies to assess the prognostic value of the identified biomarkers.

Reply: We appreciate the reviewer's suggestion. Further validation studies to assess the prognostic value of the identified biomarkers are indeed planned. However, these studies fall outside the current scope of this work, which is focused on the initial characterization of these biomarkers.

Authors could provide more data on treatment regimens and follow-up information, which could help in interpreting the prognostic significance of the biomarkers.

Reply: We acknowledge the reviewer's suggestion. However, the samples cohort included in this study does not currently have a sufficient follow-up duration to draw the conclusions requested. For breast cancer, a follow-up period of several years is typically required to gather reliable information on patient survival or response to specific treatments. We are committed to continuing this investigation and will publish updated data as soon as a sufficient follow-up period has been reached.

Authors should discuss the underlying mechanisms related to biomarker expression and chemotherapy response to gain insights into the clinical relevance of the results.

Reply: According to the reviewer’s suggestion, the discussion paragraph has been substantially revised. In the updated version of our manuscript, we provide a more concise and focused discussion, with greater emphasis on the mechanisms and potential clinical applications of our findings.

Round 2

Reviewer 1 Report

Comments and Suggestions for Authors

Some paragraphs are very short; please combine the shorter paragraphs into a single paragraph.

Author Response

Manuscript ID ijms-3412935

" Implications of Mineralization Biomarkers in Breast Cancer Outcomes Beyond Calcifications"

Submitted to: International Journal of Molecular Scinece

ROUND#3

POINT-TO-POINT REBUTTAL TO REVIEWER COMMENTS

General Comments to Editor and Reviewers

We appreciated the thoughtful and constructive criticisms and suggestions of Reviewers. His/her comments on how to improve the manuscript, which has been revised accordingly. We also appreciate the Editors for calling for a new re-submission of an improved version of our manuscript.

REVIEWER#1

Some paragraphs are very short; please combine the shorter paragraphs into a single paragraph.

Reply: Done

Reviewer 3 Report

Comments and Suggestions for Authors

The authors have revised the manuscript in accordance with the suggestions provided. I wholeheartedly recommend the publication of this article.

Author Response

REVIEWER#3

The authors have revised the manuscript in accordance with the suggestions provided. I wholeheartedly recommend the publication of this article.

Reply: We sincerely thank the reviewer for their valuable suggestions, which have significantly improved the quality of our manuscript. We appreciate your thoughtful feedback on our study.

Reviewer 4 Report

Comments and Suggestions for Authors

Accept in present form

Author Response

REVIEWER#4

Accept in present form

Reply: We sincerely thank the reviewer for their valuable suggestions, which have significantly improved the quality of our manuscript. We appreciate your thoughtful feedback on our study.